

# Entanglement dynamics in Rule 54:
# Exact results and quasiparticle picture

**Katja Klobas⋆ and Bruno Bertini**

Rudolf Peierls Centre for Theoretical Physics, Oxford University,
Parks Road, Oxford OX1 3PU, United Kingdom

⋆ katja.klobas@physics.ox.ac.uk

## Abstract

We study the entanglement dynamics generated by quantum quenches in the quantum cellular automaton Rule 54. We consider the evolution from a recently introduced class of *solvable* initial states. States in this class relax (locally) to a one-parameter family of Gibbs states and the thermalisation dynamics of local observables can be characterised exactly by means of an evolution in space. Here we show that the latter approach also gives access to the entanglement dynamics and derive exact formulas describing the asymptotic linear growth of all Rényi entropies in the thermodynamic limit and their eventual saturation for finite subsystems. While in the case of von Neumann entropy we recover exactly the predictions of the quasiparticle picture, we find no physically meaningful quasiparticle description for other Rényi entropies. Our results apply to both homogeneous and inhomogeneous quenches.

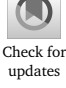

# 1   Introduction

The growth of entanglement is arguably the most universal phenomenon observed so far in studies of quantum many-body dynamics. Whenever a quantum many body system with short-range interactions is prepared in a non-equilibrium state with low entanglement and then let to evolve unitarily, the entanglement among neighbouring spatial regions is observed to grow linearly in time. For instance, this behaviour has been reported in conformal field theories, both rational [1] and holographic [2], in free systems of fermions [3] and bosons [4], as well as in interacting integrable [4,5] and non-integrable systems [6–11]. Remarkably, the growth of entanglement has even been measured in cold-atom experiments [12–14]. In essence, the only exceptions to this empirical rule are systems exhibiting localisation [15–17], confinement [18], or when the dynamics is not purely unitary, for example if the evolution is monitored with measurements [19–21].

Given such a universal phenomenology a natural direction for the theoretical research has been to find an equally universal description and identify the seemingly very general emergent laws describing it. Recent years have witnessed important progress in this direction with the proposal of two alternative effective descriptions of the spreading of entanglement which are believed to work in integrable and chaotic systems respectively. The first, known as the *quasiparticle picture* [1], explains the growth of entanglement by imagining that correlations are transported by quasiparticle excitations. These excitations, stable because of integrability, are created when the system is driven out of equilibrium and are correlated with those created nearby. During the evolution correlated quasiparticles move far apart, effectively spreading correlations and entanglement throughout the system. The second effective description is known as the *membrane picture* [22] and interprets the entanglement geometrically. In essence it claims that the entanglement between two complementary regions is given by the tension of the minimal spacetime surface that separates the two.

A quantitative verification of these pictures and their predictive power in genuinely interacting systems, however, has proven to be a daunting task. This is ultimately due to the fact that the out-of-equilibrium dynamics of interacting many-body quantum systems are generically too complicated to be characterised analytically and, moreover, the growth of entanglement provides a great limitation to the most efficient numerical methods at our disposal to treat quantum many-body systems [23]. For this reason, the benchmark provided by exact solutions in minimal solvable cases is of rare value.

Surprisingly, such a benchmark has recently become available in the case of quantum chaotic systems with the discovery of dual-unitary circuits [24]. In these systems one can exploit a duality between space and time to compute exactly the time-evolution of many relevant quantities [9–11,25–30], including that of entanglement, by performing an evolution in space (or in the "time-channel") rather than in time. Up to very recently, however, no such solvable benchmark was known for the case of interacting integrable models. The situation changed recently, when Ref. [31] presented an exact characterisation of the growth of entanglement in the quantum cellular automaton Rule 54, which can be considered one of the simplest examples of interacting integrable models (see also [32–38]). The result was again based on a time-channel approach and lead to an exact characterisation of the growth of entanglement

when the system is initialised in a particular class of initial states.

The objective of this paper is to extend the exact results presented in Ref. [31] to a larger class of initial states. This is the second of two papers dedicated to this task. While in the first part of our work [39], which in the following we will refer to as "Paper I", we focussed on the dynamics of local observables, here we consider the evolution of the entanglement. The extension that we present bares a remarkable physical significance. Indeed, while the states considered in Ref. [31] all relax (locally) to the Gibbs state with infinite temperature, here we show that exact results can be obtained also for states relaxing to richer Gibbs ensembles (characterised by an arbitrary chemical potential). This allows us, for instance, to study exactly inhomogeneous quenches giving rise to a non-trivial (generalised) hydrodynamic regime at late times [40, 41]. We use our exact results to test the predictions of the quasiparticle picture for the von Neumann entanglement entropy, both for homogeneous [5] and inhomogeneous [42, 43] quenches, providing what is, to the best of our knowledge, the first exact confirmation of this picture in the presence of either inhomogeneity or interactions. Using our exact results we also argue that no consistent quasiparticle picture can be designed in the case of Rényi entropies.

The rest of the paper is organised as follows. In Sec. 2 we introduce the time-channel approach to the entanglement dynamics in generic systems. In Sec. 3 we specialise the treatment to the case of Rule 54 and recall some of the results of Paper I that are necessary for our discussion. Sec. 4 contains the derivation of our main results, i.e. exact formulae for the stationary values eventually approached by the entropies of finite regions and for the rate of entanglement entropies after a quench from a solvable state. In Sec. 5 we derive the predictions of the quasiparticle picture for the cases of interest and compare them with our findings. Finally Sec. 6 contains our conclusions. Some more technical points and proofs are reported in the two appendices.

## 2 Entanglement dynamics in the time-channel

In this section we show that the time-channel description of the dynamics introduced in [44, 45] can also be applied to study of entanglement. As discussed in the aforementioned references (see also Paper I), this approach is based on the simple idea of evolving a many-body system in space, rather than in time, and can be applied whenever the time-evolution operator is represented as a matrix product operator (MPO). This approach has a very general scope, since essentially any evolution generated by a short-range Hamiltonian can be efficiently represented by a unitary MPO [46, 47], but it does not generically give a computational advantage. On the contrary, in certain special cases it leads to exact results. In particular, concerning the entanglement dynamics, it provides exact results in dual-unitary quantum circuits [9–11, 24] and in Rule 54 [31].

To describe the main ideas let us consider the setting described in Paper I: $2L$ qudits (with $d$ internal states) are arranged along a one-dimensional chain and driven out of equilibrium through a standard quantum quench protocol [48, 49]. Specifically, we prepare the system in a two-site shift invariant product state denoted by $|\Psi_0\rangle$ (note that here, differently from Paper I, we do not consider more general matrix product states) and evolve it with a unitary MPO (with bond dimension $\chi^2$), which we indicate by $\mathbb{U}$. The regime of interest is $L \gg t$ and we will eventually take the thermodynamic limit $L \to \infty$.

Making use of the graphical representation introduced in Paper I we depict initial state and time-evolution operator as follows

$$|\Psi_0\rangle = \underbrace{\text{⊲ ▷ ⊲ ▷ ⊲ ▷ ⊲ ▷ ⊲ ▷ ⊲ ▷}}_{2L}, \tag{1}$$

$$\mathbb{U} = \underbrace{\qquad\qquad\qquad}_{2L}, \tag{2}$$

where we assumed periodic boundary conditions and, for the time being, the tensors

$$\alpha \;—\!\!\bigcirc\!\!—\; \beta \;,\qquad \alpha \;—\!\!\bullet\!\!—\; \beta \;,\qquad \text{⊲}\;,\quad \text{▷}\;,\qquad r,s = 1,\dots,d,\quad \alpha,\beta = 1,\dots,\chi\,, \tag{3}$$

can be considered generic (the only constraints on them are that $\mathbb{U}$ must be unitary and $|\Psi_0\rangle$ normalised). As discussed in Paper I, two remarks are in order at this point: (i) here we are interested in MPOs describing local interactions and hence we should impose additional constraints on (3). However, since the upcoming discussion does not rely upon these constrains, we ignore them for the sake of simplicity; (ii) the space-time staggering in (1) and (2) is inessential and can be easily removed by appropriately merging tensors and local sites. Nevertheless, here we keep it because it arises naturally in Rule 54 which is the case of interest in this paper.

As a result of the unitary evolution, the state

$$|\Psi_t\rangle = \mathbb{U}^t |\Psi_0\rangle\,, \tag{4}$$

becomes increasingly more entangled as time advances. The growth of entanglement between a finite region $A$ and the rest of the system is quantitatively characterised by the *Rényi entropies*

$$S_A^{(\alpha)}(t) = \frac{1}{1-\alpha}\log\big[\mathrm{tr}\big(\rho_A^\alpha(t)\big)\big], \qquad \alpha \in \mathbb{R}\,, \tag{5}$$

where $\rho_A(t)$ is the density matrix of the system reduced to the subsystem $A$. In particular, the limit $\alpha \to 1$ of (5) gives the *von Neumann entropy* or *entanglement entropy*

$$S_A(t) = -\mathrm{tr}\big[\rho_A(t)\log\rho_A(t)\big], \tag{6}$$

which is the standard measure of bipartite entanglement for pure states [50]. The latter, however, is not the only interesting member of the family. Although Rényi entropies for $\alpha \neq 1$ are not entanglement measures in the strict sense, they are attracting increasing attention. This is because they characterise the spectrum of $\rho_A(t)$ — the *entanglement spectrum* — which contains non-trivial information about the system [51] (e.g. on its topological properties [52]). Moreover, and perhaps more importantly, they have recently become experimentally accessible [12–14, 53–55]. Even though physically very relevant, these quantities are notoriously hard to compute. This is especially true when considering interacting integrable models, where, up to very recently [31], they could be accessed only in the limit $t \gg A$. Indeed, as pointed out in Ref. [56], in this limit one can assume that the state of the subsystem $A$ is described by a generalised Gibbs ensemble and compute the Rényi entropies using thermodynamic Bethe ansatz (TBA) [57, 58]. The goal of this section is to derive an alternative representation for these quantities, which, as we will see, for Rule 54 allows us to access the regime $t < A$.

Let us start by looking more closely at the expression (5). Employing the diagrammatic representation in (1) and (2), we can depict the reduced density matrix at time $t$ as

$$\rho_A(t) = \text{tr}_{\bar{A}}[\mathbb{U}^t \,|\Psi_0\rangle\langle\Psi_0|\,\mathbb{U}^{-t}] = \qquad\qquad\qquad ,$$

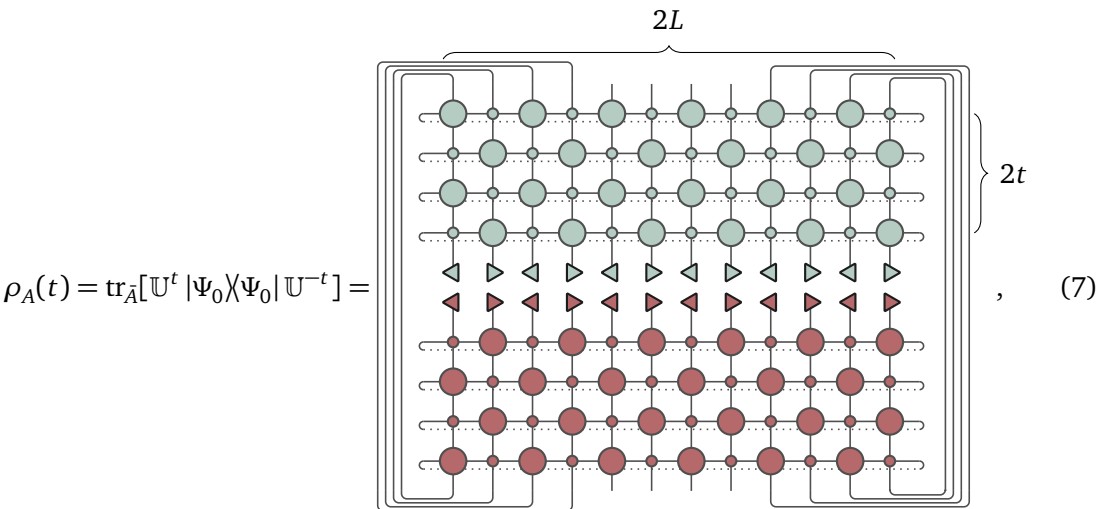

(7)

where we introduced the symbols

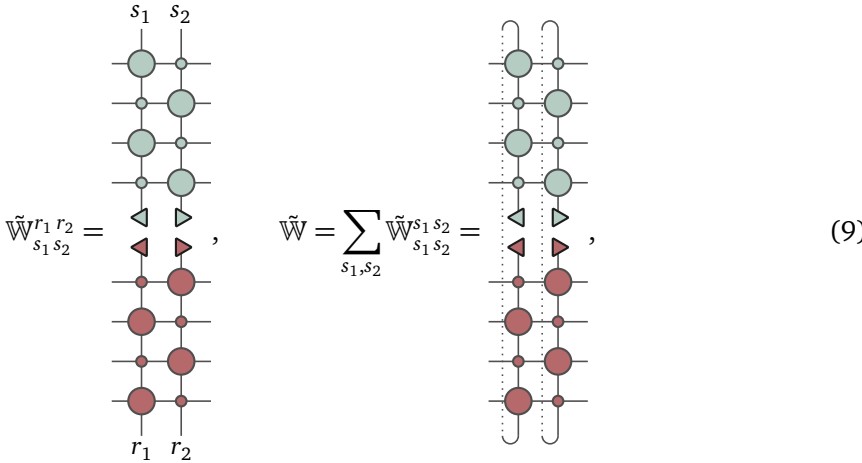

(8)

for the complex conjugate of the tensors (24).

We now interpret the tensor network (7) as the result of an evolution in space rather than in time. Specifically, by defining the space transfer matrices

$$\tilde{\mathbb{W}}^{r_1 r_2}_{s_1 s_2} = \qquad , \qquad \tilde{\mathbb{W}} = \sum_{s_1,s_2} \tilde{\mathbb{W}}^{s_1 s_2}_{s_1 s_2} = \qquad , \qquad (9)$$

we can express the reduced density matrix (7) as the following MPO

$$\rho_A(t) = \sum_{s_j, r_j \in \{0,1\}} \text{tr}\left( \tilde{\mathbb{W}}^{r_1 r_2}_{s_1 s_2} \cdots \tilde{\mathbb{W}}^{r_{|A|-1} r_{|A|}}_{s_{|A|-1} s_{|A|}} \tilde{\mathbb{W}}^{L-|A|/2} \right) |s_1 s_2 \ldots s_{|A|}\rangle\langle r_1 r_2 \ldots r_{|A|}|, \qquad (10)$$

where we conveniently consider the case of $|A|$ even.

Inserting (10) in the definition (5) of Rényi entropies and taking $\alpha = n$, with $n > 1$ integer, we have

$$S_A^{(n)}(t) = \frac{1}{1-n} \log \text{tr}\left[ (\tilde{\mathbb{W}}^{* \otimes n})^{|A|/2} \mathcal{S}_{2n}^{\dagger} (\tilde{\mathbb{W}}^{\otimes n})^{L-|A|/2} \mathcal{S}_{2n} \right], \qquad (11)$$

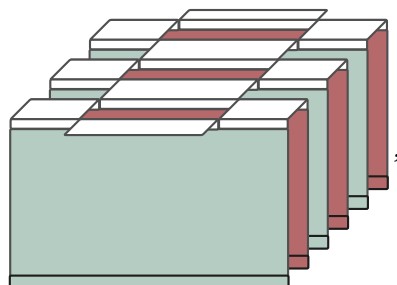

Figure 1: Pictorial representation of $\mathrm{tr}\big[\rho_A(t)^3\big]$ in the folded picture. The portions corresponding to $A$ and $\bar{A}$ are connected together in a "staggered" fashion: this staggering is implemented the operators $\mathcal{S}_{2n}$ and $\mathcal{S}_{2n}^\dagger$ in (11).

where the operator $\mathcal{S}_{2n}$ denotes a periodic shift by one in the space of the $2n$ replicas (of which $n$ correspond to forward (green) and $n$ to backward (red) time-sheets). More precisely, $\mathcal{S}_{2n}$ acts on the tensor product of $2n$ copies of the qudit chain in the $t$ direction as follows

$$\mathcal{S}_{2n}|\boldsymbol{i}_1\rangle \otimes |\boldsymbol{i}_2\rangle \otimes \cdots \otimes |\boldsymbol{i}_{2n-1}\rangle \otimes |\boldsymbol{i}_{2n}\rangle = |\boldsymbol{i}_2\rangle \otimes |\boldsymbol{i}_3\rangle \otimes \cdots \otimes |\boldsymbol{i}_{2n}\rangle \otimes |\boldsymbol{i}_1\rangle \,, \qquad \boldsymbol{i}_j \in \mathbb{Z}_d^{\times 2t}, \qquad (12)$$

where $\{|\boldsymbol{i}\rangle\}$ is a basis of $\mathbb{C}^{d^{2t}}$, which is the Hilbert space of a qudit chain of length $2t$. The operator $\mathcal{S}_{2n}$ appears because the sites of $A$ and $\bar{A}$ are contracted to different replicas in the calculation of the Rényi entropies, see Fig. 1.

To simplify (11) further we use a simple property of $\tilde{\mathbb{W}}$ (which is a special case of Property 1 in Paper I).

**Property 1.** *Whenever the initial state is normalised and the time evolution operator unitary, the spectrum of $\tilde{\mathbb{W}}$ in (9) is given by $\{0, 1\}$ and the algebraic and geometric multiplicity of the eigenvalue $1$ are equal to one.*

*Proof.* The unitarity of time-evolution implies

$$1 = \langle \Psi_t | \Psi_t \rangle = \mathrm{tr}\big[\tilde{\mathbb{W}}^L\big] = \sum_j \lambda_j^L, \qquad (13)$$

where the second equality follows directly from the definition (9) and $\lambda_j$ are eigenvalues of $\tilde{\mathbb{W}}$. This equality holds for any $L$, therefore $\lambda_j \in \{0, 1\}$ and both the geometric and algebraic multiplicity of the eigenvalue 1 have to be 1. $\square$

An immediate consequence of this is that we can write the thermodynamic limit of (11) as

$$S_{A,\mathrm{th}}^{(n)}(t) := \lim_{L\to\infty} S_A^{(n)}(t) = \frac{1}{1-n} \log\left[ \frac{{}_n\langle L|\mathcal{S}_{2n}(\tilde{\mathbb{W}}^{*\otimes n})^{|A|/2}\mathcal{S}_{2n}^\dagger|R\rangle_n}{{}_n\langle L|R\rangle_n} \right], \qquad (14)$$

with

$$_n\langle L| = {}^{n\otimes}\langle L| \,, \qquad |R\rangle_n = |R\rangle^{\otimes n} \,, \qquad (15)$$

and $\langle L|$ and $|R\rangle$ respectively denote the left and right *fixed points* (i.e. eigenvectors corresponding to the eigenvalue 1) of $\tilde{\mathbb{W}}$. We see that, in the thermodynamic limit, the $n$-th Rényi entropy is expressed as a matrix element between $n$ copies of left and right fixed points. This expression further simplifies if one considers the entanglement of half of the system

$$\lim_{|A|\to\infty} S_{A,\mathrm{th}}^{(n)}(t) = \frac{2}{1-n}\log\left[ \left| \frac{{}_n\langle L|\mathcal{S}_{2n}|R^*\rangle_n}{{}_n\langle L|R\rangle_n} \right| \right]. \qquad (16)$$

Here we introduced the shorthand notation $|R^*\rangle = |R\rangle^*$ to denote the complex conjugate of $|R\rangle$ (similarly, $\langle L^*| = \langle L|^*$) and we implicitly used

$$
{}_n\langle L^*|\mathcal{S}_{2n}^\dagger|R\rangle_n = \left({}_n\langle L|\mathcal{S}_{2n}|R^*\rangle_n\right)^*,
\tag{17}
$$

which follows directly from the permutation symmetry of $|R\rangle_n$ and ${}_n\langle L|$. Equation (16) implies that the information about the asymptotic growth of entanglement is entirely encoded in the fixed points. Note that in systems with a strict maximal speed $v_{\max}$ for the propagation of signals — as it is the case for local quantum circuits — one does not need to consider the limit $|A| \to \infty$ to obtain the simplified form (16): it is sufficient to take $|A| > 2v_{\max}t$ so that the two boundaries are causally disconnected.

This approach can also be applied when the initial state is not homogeneous (i.e. invariant under a small number of shifts), but is composed by the junction of two *different* homogeneous pieces. Namely

$$
|\Psi_0\rangle = \overbrace{\substack{L}}^{L} \quad \overbrace{\substack{R}}^{L},
\tag{18}
$$

where we took $L$ even. Quantum quenches from this kind of states are known as *bipartitioning protocols* [59–62] and can be thought of as the sudden junction of two homogeneous leads prepared in different states. In this case, the precise expression for the Rényi entropies depends on the position of $A$ with respect to the junction. For example, if the subsystem $A$ is starting at the site $x \geq 0$ (i.e. on the right of the junction), we have

$$
S_{x,A,\text{th}}^{(n)}(t) = \frac{1}{1-n}\log\left[\frac{{}_n\langle L_{\text{L}}|(\tilde{\mathbb{W}}_{\text{R}}^{\otimes n})^{x/2}\mathcal{S}_{2n}(\tilde{\mathbb{W}}_{\text{R}}^{*\otimes n})^{|A|/2}\mathcal{S}_{2n}^\dagger|R_{\text{R}}\rangle_n}{{}_n\langle L_{\text{L}}|R_{\text{R}}\rangle_n}\right].
\tag{19}
$$

For the sake of simplicity in this paper we only consider the special case $x = 0$, i.e. when $A$ starts right at the junction, i.e.

$$
S_{A,\text{th}}^{(n)}(t) = \frac{1}{1-n}\log\left[\frac{{}_n\langle L_{\text{L}}|(\mathcal{S}_{2n}(\tilde{\mathbb{W}}_{\text{R}}^{*\otimes n})^{|A|/2}\mathcal{S}_{2n}^\dagger|R_{\text{R}}\rangle_n}{{}_n\langle L_{\text{L}}|R_{\text{R}}\rangle_n}\right].
\tag{20}
$$

If, in addition, the system has a strict maximal velocity, and the subsystem large enough, $|A| > 2v_{\max}t$, the above expression reduces to

$$
S_{A,\text{th}}^{(n)}(t) = \frac{1}{1-n}\log\left[\frac{{}_n\langle L_{\text{L}}|\mathcal{S}_{2n}|R_{\text{R}}^*\rangle_n}{{}_n\langle L_{\text{L}}|R_{\text{R}}\rangle_n}\right] + \frac{1}{1-n}\log\left[\left(\frac{{}_n\langle L_{\text{R}}|\mathcal{S}_{2n}|R_{\text{R}}^*\rangle_n}{{}_n\langle L_{\text{R}}|R_{\text{R}}\rangle_n}\right)^*\right].
\tag{21}
$$

Note that the two terms on the r.h.s. can be directly interpreted as the entanglement produced at the two boundary points between $A$ and $\bar{A}$. Indeed, the second term depends on the parameters of the right lead only, while the first depends on the parameters of both left and right lead. Consistently, repeating the same construction in the case of open boundary conditions and taking $A$ to be semi-infinite one finds [31]

$$
\lim_{|A|\to\infty} S_{A,\text{th}}^{(n)}(t)\Big|_{\text{obc}} = \frac{1}{1-n}\log\left[\frac{{}_n\langle L_{\text{L}}|\mathcal{S}_{2n}|R_{\text{R}}^*\rangle_n}{{}_n\langle L_{\text{L}}|R_{\text{R}}\rangle_n}\right].
\tag{22}
$$

Indeed, in this case there is a single boundary point between $A$ and $\bar{A}$.

Our main goal will be to exploit the representations (16) and (21) to find the asymptotic behaviour of Rényi entropies for large times. Since (21) reduces to (14) for

$$
\substack{\triangleleft \\ R} = \substack{\triangleleft \\ L} = \triangleleft, \qquad \substack{\triangleright \\ R} = \substack{\triangleright \\ L} = \triangleright,
\tag{23}
$$

we can, without loss of generality, consider the inhomogeneous case (21) only.

# 3  A solvable case: quantum cellular automaton Rule 54

The practical convenience of the representation (21) depends on the form of the fixed points $\langle L|$ and $|R\rangle$. For instance, they become extremely useful when the fixed points are written as matrix product states (MPS)s with a constant (i.e. time independent) bond dimension. This kind of simplification arises for some particular choices of the tensors (3), i.e. for particular systems and initial states [10, 24, 31].

Here we focus on one of such choices. Specifically, we consider the quantum cellular automaton Rule 54, originally introduced in Ref. [63], which has been recently shown to offer an exactly solvable benchmark for interacting integrable many-body dynamics, both in the classical [64–70], and quantum [31, 71–75] realm (see also the recent review [76]). We can interpret it as a local quantum circuit where the time-evolution operator is written in the form (2) with tensors [31, 70]

$$
\alpha \overset{r}{\underset{s}{\text{—}\bigcirc\text{—}}} \beta = \delta_{\chi(s,\beta,r),\alpha}, \qquad \alpha \overset{r}{\underset{s}{\text{—}\mid\text{—}}} \beta = \delta_{s,\beta}\delta_{\beta,r}\delta_{r,\alpha}, \tag{24}
$$

where $d = \chi = 2$ and $\chi(s,\beta,r) = (s + \beta + r + sr)\bmod 2$. Note that, since Rule 54 can be represented as a local quantum circuit, is has a strict maximal velocity $v_{\max}$ for the propagation of signals. In particular, for our choice of units we have $v_{\max} = 2$.

Next, we consider initial-states

$$
\left|\Psi_{\vartheta,\varphi}\right\rangle = \underbrace{\downarrow\,\downarrow\,\downarrow\,\downarrow\,\downarrow\,\downarrow\,\downarrow\,\downarrow}_{\vartheta} \tag{25}
$$

with tensors of the form

$$
\overset{s}{\underset{\triangleleft}{\downarrow}} = e^{i\varphi_1}\delta_{s,0}, \qquad \overset{s}{\underset{\vartheta}{\downarrow}}_{\triangleright} = \sqrt{1-\vartheta}\,\delta_{s,0} + \sqrt{\vartheta}\,e^{i\varphi_2}\delta_{s,1}, \tag{26}
$$

where the parameter $\vartheta \in [0,1]$ will be referred to as the *filling* while $\varphi_{1/2} \in [0,2\pi]$ as the *phases*.

In Paper I we prove that, choosing tensors of the form (24) and (26), the fixed points, $\langle R|$ and $|L\rangle$, are MPSs of bond dimension 3. The latter depend on the filling but are independent of the phases. In fact, one can prove that fixed points are the same also when choosing different phases at each spatial point (as long as $\vartheta$ is the same everywhere). Explicitly, we have

$$
\langle L_{\vartheta}| = \left\| \begin{matrix} \end{matrix} \right\|, \qquad |R_{\vartheta}\rangle = \left\| \begin{matrix} \end{matrix} \right\|, \tag{27}
$$

where the "bulk" tensors are given by

$$
0\!-\!\diamondsuit_\vartheta\!-\!0 = \begin{bmatrix} 1-\vartheta & 1-\vartheta & -(1-\vartheta) \\ \vartheta & \vartheta & 1-\vartheta \\ \vartheta & -\dfrac{\vartheta^2}{1-\vartheta} & -\vartheta \end{bmatrix}, \qquad 0\!-\!\diamondsuit\!-\!1 = 1\!-\!\diamondsuit\!-\!0 = \begin{bmatrix} 0 & 1-\vartheta & -(1-\vartheta) \\ \vartheta & 0 & 0 \\ \vartheta & 0 & 0 \end{bmatrix},
$$

$$
s\!-\!\blacksquare\!-\!r = \begin{bmatrix} \delta_{r,0}\delta_{s,0} & 0 & 0 \\ 0 & \delta_{r,1}\delta_{s,1} & 0 \\ 0 & 0 & \delta_{r,1}\delta_{s,1} \end{bmatrix}, \qquad\qquad 1\!-\!\diamondsuit\!-\!1 = \begin{bmatrix} 0 & 1 & 0 \\ 1 & 0 & 0 \\ 0 & 0 & 0 \end{bmatrix},
$$

$$\tag{28}$$

and boundary vectors are

$$
\blacktriangleright = \begin{bmatrix} 1 \\ 0 \\ 0 \end{bmatrix}, \qquad \blacktriangleleft_\vartheta = -\frac{1}{1-\vartheta}\begin{bmatrix} (1-\vartheta)^2 \\ \vartheta(1-\vartheta) \\ -\vartheta^2 \end{bmatrix}, \qquad \perp = \frac{1}{\sqrt{2}}\begin{bmatrix} 1 \\ 1 \\ 0 \end{bmatrix}. \tag{29}
$$

These choices give left and right fixed point fulfilling

$$
\big\langle L_{\vartheta_1}\big|R_{\vartheta_2}\big\rangle = \; \bigsqcup_{\vartheta_1} \; \bigsqcup_{\vartheta_2} \; = 1, \qquad \forall \vartheta_1, \vartheta_2. \tag{30}
$$

In the above diagrams we explicitly reported $\vartheta$ to signal the dependence on the filling. In the following, however, whenever the choice of $\vartheta$ is unambiguous we will ease the notation by removing it.

As proven in Paper I the state $\big|\Psi_{\vartheta,\varphi}\big\rangle$ relaxes (locally) to a family of Gibbs states. In particular considering density matrix reduced to a finite subsystem $A$ we have

$$
\rho_A(t) \simeq \rho_{\mathrm{GE},A} = \frac{\mathrm{tr}_{\bar{A}}(e^{-\mu(\vartheta)N})}{\mathrm{tr}(e^{-\mu(\vartheta)N})}, \qquad N = N_+ + N_-, \tag{31}
$$

where $\simeq$ denotes the leading contribution for large times, $N_\pm$ are the number of left and right-moving quasiparticles (solitons) explicitly given by [72,77]

$$
\begin{aligned}
N_+ &= \sum_{x\in\mathbb{Z}_L} P^-_{2x}P^-_{2x+1} + \sum_{x\in\mathbb{Z}_{2L}} P^+_x P^-_{x+1} P^+_{x+2}, \\
N_- &= \sum_{x\in\mathbb{Z}_L} P^-_{2x-1}P^-_{2x} + \sum_{x\in\mathbb{Z}_{2L}} P^+_x P^-_{x+1} P^+_{x+2},
\end{aligned}
\qquad P^\pm := \frac{\mathbb{1}\pm\sigma_3}{2}, \tag{32}
$$

and the chemical potential $\mu(\theta)$ reads as

$$
e^{-\mu(\vartheta)} = \frac{\vartheta}{1-\vartheta} \qquad\Rightarrow\qquad \vartheta = \frac{1}{1+e^{\mu(\vartheta)}}. \tag{33}
$$

This shows that $\vartheta$ in (25) sets the density of quasiparticles in the stationary state.

In fact, the states (25) can also be used to design solvable bipartitioning protocols. Indeed, as proven in Paper I, considering initial states of the form

$$
\big|\Psi_{\vartheta_\mathrm{L},\varphi_\mathrm{L},\vartheta_\mathrm{R},\varphi_\mathrm{R}}\big\rangle = \big|\Psi_{\vartheta_\mathrm{L},\varphi_\mathrm{L}}\big\rangle \otimes \big|\Psi_{\vartheta_\mathrm{R},\varphi_\mathrm{R}}\big\rangle, \tag{34}
$$

one finds $\langle L_\mathrm{L}| = \big\langle L_{\vartheta_\mathrm{L}}\big|$ and $|R_\mathrm{R}\rangle = \big|R_{\vartheta_\mathrm{R}}\big\rangle$ (with both $\langle L_\vartheta|$ and $|R_\vartheta\rangle$ of the form (27)). In this case any finite subsystem $A$ at finite distance from the junction relaxes to a family of *generalised Gibbs states*. Namely

$$
\rho_A(t) \simeq \rho_{\mathrm{GGE},A} = \frac{\mathrm{tr}_{\bar{A}}(e^{-\mu_\mathrm{L}(\vartheta_\mathrm{L},\vartheta_\mathrm{R})N_- -\mu_\mathrm{R}(\vartheta_\mathrm{L},\vartheta_\mathrm{R})N_+})}{\mathrm{tr}(e^{-\mu_\mathrm{L}(\vartheta_\mathrm{L},\vartheta_\mathrm{R})N_- -\mu_\mathrm{R}(\vartheta_\mathrm{L},\vartheta_\mathrm{R})N_+})}, \tag{35}
$$

where $\mu_{R/L}(\vartheta_L, \vartheta_R)$ is given by

$$e^{-\mu_{R/L}(\vartheta_L, \vartheta_R)} = \frac{\vartheta_{R/L}(1 - \vartheta_{L/R})}{(1 - \vartheta_{R/L})^2}.$$ (36)

Importantly, in Rule 54 the relaxation happens with finite rate [31] (see also Paper I). In particular, the finite-time corrections to (31) and (35) are exponentially small in $t - 3|A|/2$.

Finally, we recall (see e.g. Paper I) that $\rho_{\mathrm{GGE},A}$ in Eq. (35) (and hence also its particular case (31)) is conveniently expressed in terms of the following MPO

$$\rho_{\mathrm{GGE},A} = \frac{1}{Z_A} \overbrace{\phantom{xxxxxxxx}}^{|A|} , \qquad Z_A = 1 + \vartheta_L + \vartheta_R.$$ (37)

The bulk tensors are diagonal in the two copies of the physical space, and the auxiliary space is 3-dimensional,

$$\begin{array}{c} {}^0 \\ \blacktriangleleft \\ {}_0 \end{array} = \begin{bmatrix} 1 & 0 & 0 \\ e^{-\mu_R(\vartheta_L, \vartheta_R)} & 0 & 0 \\ 1 & 0 & 0 \end{bmatrix}, \qquad \begin{array}{cc} {}^0 & {}^1 \\ \blacktriangleleft & = \blacktriangleleft \\ {}_1 & {}_0 \end{array} = 0,$$

$$\begin{array}{c} {}^1 \\ \blacktriangleleft \\ {}_1 \end{array} = \begin{bmatrix} 0 & e^{-\mu_R(\vartheta_L, \vartheta_R)} & 0 \\ 0 & 0 & 1 \\ 0 & 0 & e^{-\mu_L(\vartheta_L, \vartheta_R)} \end{bmatrix}, \qquad \blacktriangleright = (1 - \vartheta_L)(1 - \vartheta_R) \left. \blacktriangleleft \right|_{\mu_L \leftrightarrow \mu_R}.$$ (38)

The boundary tensors are 3-dimensional (row and column) vectors and their explicit expression is reported in Appendix A of Paper I.

# 4 Exact results for Rényi entropies

In this section we show that combining the representations (21) and (22) with the exact expressions (27) the problem of computing the growth of Rényi entropies is mapped into that of contracting a certain tensor network. This can be done exactly in the asymptotic limit $1 \ll t \leq |A|/4$. Moreover, using (37), we also show that also the stationary value reached by the entropies for large times is characterised by a tensor network. As we shall see, the latter is contracted exactly in the limit of large $|A|$. Let us begin by proving the latter statement.

## 4.1 Stationary values

As a consequence of (35) we have that for $t > 3|A|/2$ the Rényi entropies fulfil

$$S_A^{(\alpha)}(t) \simeq S_{\mathrm{GGE},A}^{(\alpha)} = \frac{1}{1-\alpha} \log\left[ \mathrm{tr}\left( \rho_{\mathrm{GGE},A}^\alpha \right) \right],$$ (39)

where we recall that $\simeq$ denotes equality up to exponentially small corrections and $\rho_{\mathrm{GGE},A}$ the GGE reduced to the subsystem $A$. Using the MPO representation of $\rho_{\mathrm{GGE},A}$ (cf. Eq. (37)) we can express Rényi entropies with index $n$ (integer and larger than one) in terms of the following

tensor network

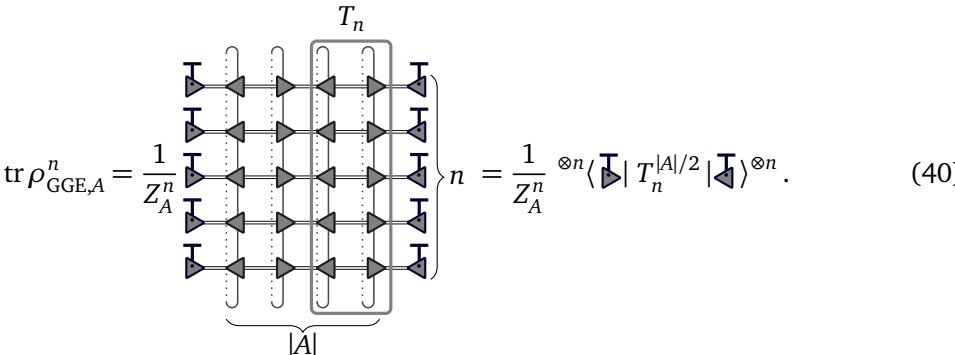

$$
\operatorname{tr}\rho_{\mathrm{GGE},A}^{n}=\frac{1}{Z_{A}^{n}}\ \cdots\ n\ =\frac{1}{Z_{A}^{n}}{}^{\otimes n}\langle\,\mathbf{\triangleright}|\ T_{n}^{|A|/2}\,|\mathbf{\triangleleft}\rangle^{\otimes n}. \tag{40}
$$

This immediately implies that for large $|A|$ the Rényi entropy is dominated by the leading eigenvalue $\Lambda_{n}$ of the transfer matrix $T_{n}$ (defined in the above diagram). Namely

$$
S_{\mathrm{GGE},A}^{(n)}=\frac{|A|}{2(1-n)}\log\left(\Lambda_{n}(\vartheta_{1},\vartheta_{2})\right)+\mathcal{O}\left(|A|^{0}\right), \tag{41}
$$

where we used the fact that $Z_{A}$ does not grow with $|A|$ (cf. Eq. (37)).

To find $\Lambda_{n}$ we make use of the following relations

$$
\cdots = \cdots , \qquad \cdots = \cdots , \tag{42}
$$

where we introduced the projector defined by

$$
\begin{matrix} x \!-\!\!\bullet\!\!-\! z \\ \ \ \Big| \\ y \!-\!\!\bullet\!\!-\! w \end{matrix} = \delta_{x,y}\delta_{y,z}\delta_{z,w}. \tag{43}
$$

The identities (42) imply that the eigenvalues of $T_{n}$ coincide with the spectrum of the reduced transfer matrix $\tilde{T}_{n}$ defined by applying the projector (43) on all the pairs of auxiliary legs

$$
\tilde{T}_{n}= \cdots . \tag{44}
$$

Therefore the non-zero eigenvalues of $T_{n}$ are given by the spectrum of a $3\times 3$ matrix

$$
\mathrm{Sp}(T_{n})=\mathrm{Sp}(\tilde{T}_{n})=\{0\}\cup\mathrm{Sp}\left(\bar{\vartheta}_{1}^{n}\bar{\vartheta}_{2}^{n}\begin{bmatrix} 1+e^{-n(\mu_{1}+\mu_{2})} & e^{-n\mu_{1}} & e^{-n\mu_{2}} \\ 1+e^{-n\mu_{2}} & e^{-n(\mu_{1}+\mu_{2})} & e^{-n\mu_{2}} \\ 1+e^{-n\mu_{1}} & e^{-n\mu_{1}} & e^{-n(\mu_{1}+\mu_{2})} \end{bmatrix}\right), \tag{45}
$$

where we use the shorthand notation $\bar{\vartheta}_{1/2}=1-\vartheta_{1/2}$. In particular, the leading eigenvalue $\Lambda_{n}(\vartheta_{1},\vartheta_{2})$ can be expressed as

$$
\Lambda_{n}(\vartheta_{1},\vartheta_{2})=\frac{(1-\vartheta_{1})^{n}(1-\vartheta_{2})^{n}}{(1-\vartheta_{1}^{(n)})(1-\vartheta_{2}^{(n)})}, \tag{46}
$$

where we defined $\vartheta_{1,2}^{(n)}$ fulfilling

$$\frac{\vartheta_{1,2}^{(n)}(1-\vartheta_{2,1}^{(n)})}{(1-\vartheta_{1,2}^{(n)})^2} = \left(\frac{\vartheta_{1,2}(1-\vartheta_{2,1})}{(1-\vartheta_{1,2})^2}\right)^n. \tag{47}$$

Note that $\vartheta_{1,2}^{(n)}$ can be understood as generalisations of the filling functions $\vartheta_{1,2}$ to the case where chemical potentials $\mu_{1,2}$ are replaced by $n\mu_{1,2}$ (see Eq. (36)). We also remark that the result obtained by substituting (46) in (41) agrees with the TBA prediction of Ref. [56].

The result (41) can be analytically continued to $\mathcal{D} = \{z \in \mathbb{C} : \text{Re}[z] > 0\}$. Indeed, the function $\Lambda_z(\vartheta_1, \vartheta_2)$ — obtained by replacing $n$ in Eq. (46) with $z \in \mathbb{C}$ — is holomorphic and bounded in $\mathcal{D}$. Therefore, Carlson's Theorem [78] ensures that it is the only analytic continuation of $\{\Lambda_n(\vartheta_1, \vartheta_2)\}_{n=1,2,3,\dots}$ fulfilling

$$|\Lambda_z(\vartheta_1, \vartheta_2)| \leq Ce^{\tau|z|}, \quad z \in \mathcal{D}, \qquad |\Lambda_{1+iy}(\vartheta_1, \vartheta_2)| \leq Ce^{c|y|}, \quad y \in \mathbb{R}, \tag{48}$$

with $C, \tau \in \mathbb{R}$ and $c < \pi$. As this is a requirement that we expect from physical grounds, we choose $\Lambda_z(\vartheta_1, \vartheta_2)$ as the relevant analytic continuation. In particular, in the limit $z \to 1$ we find

$$S_{\text{GGE},A} = -\frac{|A|}{2} \sum_{j=1}^{2} \frac{1+2\vartheta_{3-j}}{1+\vartheta_1+\vartheta_2} \left(\vartheta_j \log \vartheta_j + (1-\vartheta_j)\log(1-\vartheta_j)\right) + \mathcal{O}(|A|^0), \tag{49}$$

which coincides with the expression of the Yang-Yang entropy in the state (35) [73]. In the homogeneous case, when $\vartheta_1 = \vartheta_2 = \vartheta$, Rényi entropies take a free-fermionic form (see e.g. [56])

$$S_{\text{GE},A}^{(n)} = -\frac{|A|}{1-n} \log\left((1-\vartheta)^n + \vartheta^n\right) + \mathcal{O}(|A|^0). \tag{50}$$

## 4.2 Asymptotic slopes

Let us consider the elementary building blocks

$$b_n(\vartheta_1, \vartheta_2) := \frac{1}{1-n} \log\left[\frac{{}_n\langle L_{\vartheta_1}|\mathcal{S}_{2n}|R_{\vartheta_2}^*\rangle_n}{{}_n\langle L_{\vartheta_1}|R_{\vartheta_2}\rangle_n}\right], \tag{51}$$

where $\langle L_{\vartheta_1}|, |R_{\vartheta_2}\rangle$ are both of the form (27). Recalling (21) we see that $b_n(\vartheta_1, \vartheta_2)$ can be interpreted as the $n$-th Rényi entropy generated at one of the boundaries of the subsystem $A$. This means that, upon analytic continuation, evaluating (51) gives direct access to *all* Rényi entropies (including von Neumann) for $t \leq |A|/4$.

Considering the graphical representation (27) of the fixed points, we can express the matrix element in (51) in terms of the following tensor network

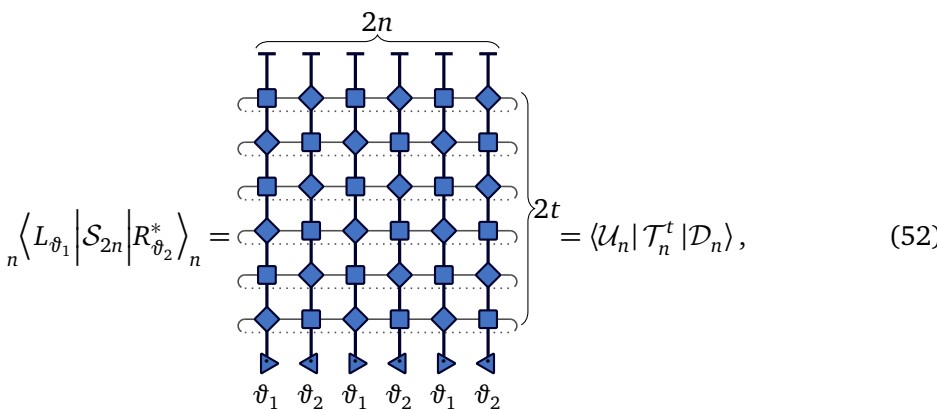

$$_n\langle L_{\vartheta_1}|\mathcal{S}_{2n}|R_{\vartheta_2}^*\rangle_n = \quad = \langle \mathcal{U}_n|\mathcal{T}_n^t|\mathcal{D}_n\rangle, \tag{52}$$

where we introduced

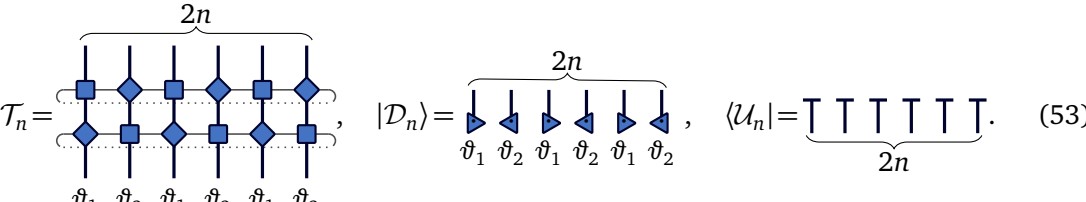

$$\mathcal{T}_n = \quad , \quad |\mathcal{D}_n\rangle = \quad , \quad \langle \mathcal{U}_n| = \quad . \tag{53}$$

From the representation (52) we see that the asymptotic behaviour of (51) is determined by the largest eigenvalue of the transfer matrix $\mathcal{T}_n$. Therefore, we proceed by identifying its spectrum. To this aim it is convenient to merge together the tensors ◆, ◼ on two consecutive rows and columns, i.e.

$$M_{ab}^{xy} = \quad \equiv \quad , \quad \begin{array}{ll} x = 3z_1 + z_2, & y = 3w_1 + w_2, & x,y \in \mathbb{Z}_9, \\ a = 2r_1 + r_2, & b = 2s_1 + s_2, & a,b \in \mathbb{Z}_4, \end{array} \tag{54}$$

so that $\mathcal{T}_n$ is rewritten in terms of $n$ horizontally connected tensors $M$ with periodic boundaries

$$\mathcal{T}_n = \quad . \tag{55}$$

Moreover, we also make a convenient local basis transformation

$$\tilde{M} = \quad = \quad , \qquad \bigcirc = P, \qquad \ominus = P^{-1}, \tag{56}$$

where we defined

$$P = \begin{bmatrix}
1 & & & & & & & & \\
& 1 & 1 & & & & & & \\
& 1 & -\dfrac{\vartheta_2}{1-\vartheta_2} & & & & & & \\
& & & 1 & & & 1 & & \\
& & & & 1 & 1 & & 1 & 1 \\
& & & & 1 & -\dfrac{\vartheta_2}{1-\vartheta_2} & & 1 & -\dfrac{\vartheta_2}{1-\vartheta_2} \\
& & & 1 & & & -\dfrac{\vartheta_1}{1-\vartheta_1} & & \\
& & -\dfrac{\vartheta_1}{1-\vartheta_1} & -\dfrac{\vartheta_1}{1-\vartheta_1} & & & 1 & 1 \\
& & -\dfrac{\vartheta_1}{1-\vartheta_1} & \dfrac{\vartheta_1\vartheta_2}{(1-\vartheta_1)(1-\vartheta_2)} & & & 1 & -\dfrac{\vartheta_2}{1-\vartheta_2}
\end{bmatrix}. \tag{57}$$

We denote by $\tilde{\mathcal{T}}_n$ the transfer matrix in the new basis (defined as (55), with $M$ replaced by $\tilde{M}$).

Since $\tilde{\mathcal{T}}_n$ and $\mathcal{T}_n$ are related by a similarity transformation, their spectra coincide. To determine them we make use of the following Lemma (proven in Appendix A).

**Lemma 1.** *For any $k \geq 1$*

$$\left(\prod_{j=1}^{k} \delta_{y_k,x_k}\right) \tag{58}$$

This lemma has the remarkable consequence that traces of powers of $\mathcal{T}_n$ can be obtained by considering a simple $9 \times 9$ matrix. Namely we have

$$\mathrm{tr}\big(\mathcal{T}_n^k\big) = \mathrm{tr}\big(\tilde{\mathcal{T}}_n^k\big) = \sum_{x_1,x_2,\dots,x_k} \qquad = \qquad k = \mathrm{tr}\big(\tilde{\tau}_n^k\big), \tag{59}$$

where we introduced the tensor

$$b \,\text{—}\!\boxed{n}\!\text{—}\, a = \big(\tilde{M}_{ab}^{xy}\big)^n . \tag{60}$$

Explicitly, the matrix elements of $\tilde{\tau}_n$ are expressed as

$$\big[\tilde{\tau}_n\big]_{x_1,x_2} = \sum_{a_1,\dots,a_n=1}^{4} \tilde{M}_{a_1 a_2}^{x_1 x_2} \cdots \tilde{M}_{a_n a_1}^{x_1 x_2}, \tag{61}$$

which yields

$$\tilde{\tau}_n = \begin{bmatrix} (1-\vartheta_1)^n(1-\vartheta_2)^n & 0 & 0 & 0 & 0 & 1 & (1-\vartheta_2)^n & 0 & 1 \\ \vartheta_2^n(1-\vartheta_1)^n & 0 & 0 & 0 & 0 & 0 & \vartheta_2^n & 0 & 0 \\ 0 & 0 & 0 & 0 & 1 & 0 & 0 & 1 & 0 \\ 0 & \vartheta_1^n & \vartheta_1^n & 0 & 0 & 0 & 0 & 0 & 0 \\ 0 & 0 & 0 & \vartheta_2^n & 0 & 0 & 0 & 0 & 0 \\ 0 & 0 & 0 & (1-\vartheta_2)^n & 0 & 0 & 0 & 0 & 0 \\ 0 & (1-\vartheta_1)^n & (1-\vartheta_1)^n & 0 & 0 & 0 & 0 & 0 & 0 \\ \vartheta_1^n \vartheta_2^n & 0 & 0 & 0 & 0 & 0 & 0 & 0 & 0 \\ \vartheta_1^n(1-\vartheta_2)^n & 0 & 0 & 0 & 0 & 0 & 0 & 0 & 0 \end{bmatrix} . \tag{62}$$

Since the relation (59) holds for any $k$, the non-zero eigenvalues of $\mathcal{T}_n$ and $\tilde{\tau}_n$ coincide. The eigenvalues of the latter are easily obtained. In particular, it is straightforward to see that the only three non-zero eigenvalues of $\tilde{\tau}_n$ are the solutions to the following cubic equation

$$\lambda^3 = \big((1-\vartheta_1)^n\lambda + \vartheta_1^n\big)\big((1-\vartheta_2)^n\lambda + \vartheta_2^n\big). \tag{63}$$

The main properties of this equation are studied in Appendix B and can be summarised as follows

**Lemma 2.** *For $\vartheta_1, \vartheta_2 \in (0,1)$ the solution of Eq. (63) with strictly larger magnitude is real and positive. Its explicit expression reads as*

$$\lambda_n(\vartheta_1,\vartheta_2) = \frac{(1-\vartheta_1)^n(1-\vartheta_2)^n}{3} + \sqrt[3]{\Delta_{n,1} + \sqrt{\Delta_{n,1}^2 - \Delta_{n,2}^3}} + \frac{\Delta_{n,2}}{\sqrt[3]{\Delta_{n,1} + \sqrt{\Delta_{n,1}^2 - \Delta_{n,2}^3}}}, \tag{64}$$

*where*

$$\Delta_{n,1} = \frac{1}{6}(1-\vartheta_1)^{2n}(1-\vartheta_2)^{2n}\sum_{j=1}^{2}\left(\frac{\vartheta_j}{1-\vartheta_j}\right)^n + \frac{1}{2}\vartheta_1^n\vartheta_2^n + \frac{1}{27}(1-\vartheta_1)^{3n}(1-\vartheta_2)^{3n},$$
$$\Delta_{n,2} = \frac{1}{3}\vartheta_1^n(1-\vartheta_2)^n + \frac{1}{3}\vartheta_2^n(1-\vartheta_1)^n + \frac{1}{9}(1-\vartheta_1)^{2n}(1-\vartheta_2)^{2n}. \tag{65}$$

Putting all together and noting that ${}_n\langle L_{\vartheta_1}|R_{\vartheta_2}\rangle_n = 1$ (cf. (30)) we find that for $t \gg 1$, the building block (51) displays a linear growth with slope given by

$$r_n(\vartheta_1,\vartheta_2) := \lim_{t\to\infty}\frac{b_n(\vartheta_1,\vartheta_2)}{t} = \frac{1}{1-n}\log[\lambda_n(\vartheta_1,\vartheta_2)]. \tag{66}$$

For example, in the case of the min-entropy (i.e. $n \to \infty$) we find the following explicit result

$$r_\infty(\vartheta_1,\vartheta_2) = \begin{cases} -\log[(1-\vartheta_1)(1-\vartheta_2)], & \begin{aligned}\frac{\vartheta_1}{1-\vartheta_1} &\le (1-\vartheta_1)(1-\vartheta_2), \\ \frac{\vartheta_2}{1-\vartheta_2} &\le (1-\vartheta_1)(1-\vartheta_2),\end{aligned} \\[2ex] -\log[\vartheta_1^{1/2}(1-\vartheta_2)^{1/2}], & \begin{aligned}\frac{\vartheta_1}{1-\vartheta_1} &> \vartheta_1^{1/2}(1-\vartheta_2)^{1/2}, \\ \frac{\vartheta_2}{1-\vartheta_2} &\le \vartheta_1^{1/2}(1-\vartheta_2)^{1/2},\end{aligned} \\[2ex] -\log[\vartheta_2^{1/2}(1-\vartheta_1)^{1/2}], & \begin{aligned}\frac{\vartheta_2}{1-\vartheta_2} &> \vartheta_2^{1/2}(1-\vartheta_1)^{1/2}, \\ \frac{\vartheta_1}{1-\vartheta_1} &\le \vartheta_2^{1/2}(1-\vartheta_1)^{1/2},\end{aligned} \\[2ex] -\log[\vartheta_1^{1/3}\vartheta_2^{1/3}], & \begin{aligned}\frac{\vartheta_1}{1-\vartheta_1} &\ge \vartheta_1^{1/3}\vartheta_2^{1/3}, \\ \frac{\vartheta_2}{1-\vartheta_2} &\ge \vartheta_1^{1/3}\vartheta_2^{1/3}.\end{aligned} \end{cases} \tag{67}$$

Note that the above characterisation of the spectrum of $\mathcal{T}_n$ can also be used to find the leading corrections to (66). To do that, however, one would also need to find the eigenvector associated to $\lambda_n(\vartheta_1,\vartheta_2)$.

Substituting (66) in (21) we arrive at the main result of this paper

$$\lim_{\substack{t\to\infty \\ |A|/t=\zeta\ge 4}}\frac{S_{A,\text{th}}^{(n)}(t)}{t} = r_n(\vartheta_\text{L},\vartheta_\text{R}) + r_n(\vartheta_\text{R},\vartheta_\text{R}). \tag{68}$$

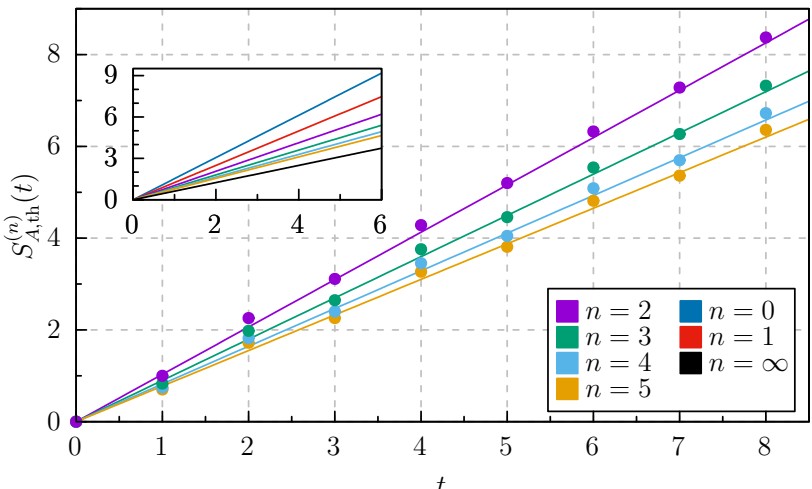

Figure 2: Growth of the Rényi $n$-entropy after a homogeneous quench from the state characterised by $(\vartheta_{\mathrm{L}}, \vartheta_{\mathrm{R}}) = (0.65, 0.15)$, for typical values of $n$. Solid lines are the asymptotic prediction obtained by neglecting the subleading corrections and taking into account only the largest eigenvalue of the tensor network (52), while the dots correspond to exact finite-time values in the $L \to \infty$ limit.

This equation provides a rigorous proof of the fact that Rényi entropies of large subsystems grow linearly in the asymptotic regime and gives an exact expression for their slope. A comparison between the asymptotic result (68) and the exact numerical evaluation of (21) for finite times is reported in Fig. 2.

We recall that (68) applies to the case of a bipartitioning protocol with two leads initially prepared in different solvable states (25) and where the subsystem $A$ starts at the junction. The special case $\vartheta_{\mathrm{L}} = \vartheta_{\mathrm{R}} = \vartheta$ describes the growth of entanglement after a homogeneous quench from a solvable state. A remarkable consequence of (68) is that the entanglement velocity

$$v_n^{\mathrm{E}}(\vartheta_{\mathrm{L}}, \vartheta_{\mathrm{R}}) := \lim_{\substack{t \to \infty \\ |A|/t=\zeta \geq 4}} \frac{S_{A,\mathrm{th}}^{(n)}(t)}{t s_{\mathrm{GGE}}^{(n)}} = \frac{r_n(\vartheta_{\mathrm{L}}, \vartheta_{\mathrm{R}})}{s_{\mathrm{GGE}}^{(n)}} + \frac{r_n(\vartheta_{\mathrm{R}}, \vartheta_{\mathrm{R}})}{s_{\mathrm{GGE}}^{(n)}}, \tag{69}$$

where $s_{\mathrm{GGE}}^{(n)}$ is the entropy density of the GGE (cf. (41)), depends non-trivially on $n$. See Fig. 3 for a representative example.

The result (66) can again be analytically continued to $\mathcal{D} = \{z \in \mathbb{C} : \mathrm{Re}[z] > 0\}$. Indeed, the function $\lambda_z(\vartheta_1, \vartheta_2)$ — obtained by replacing $n$ in (64) with $z \in \mathbb{C}$ — is holomorphic and bounded in $\mathcal{D}$. Specifically (see Appendix B)

$$|\lambda_z(\vartheta_1, \vartheta_2)| \leq \lambda_{\mathrm{Re}[z]}(\vartheta_1, \vartheta_2) < 3. \tag{70}$$

Applying again Carlson's Theorem [78] we then have that $\lambda_z(\vartheta_1, \vartheta_2)$ is the only analytic continuation of $\{\lambda_n(\vartheta_1, \vartheta_2)\}_{n=1,2,3,\ldots}$ which fulfils the physically sensible bounds

$$|\lambda_z(\vartheta_1, \vartheta_2)| \leq C e^{\tau|z|}, \quad z \in \mathcal{D}, \qquad |\lambda_{1+iy}(\vartheta_1, \vartheta_2)| \leq C e^{c|y|}, \quad y \in \mathbb{R}, \tag{71}$$

with $C, \tau \in \mathbb{R}$ and $c < \pi$.

Considering now $z = 1 + \delta$ with $\delta \ll 1$ from (64) we find

$$\lambda_{1+\delta}(\vartheta_1, \vartheta_2) = 1 + \frac{\delta}{1 + \vartheta_1 + \vartheta_2} \sum_{j=1}^{2} \big(\vartheta_j \log \vartheta_j + (1 - \vartheta_j) \log(1 - \vartheta_j)\big) + \mathcal{O}(\delta^2), \tag{72}$$

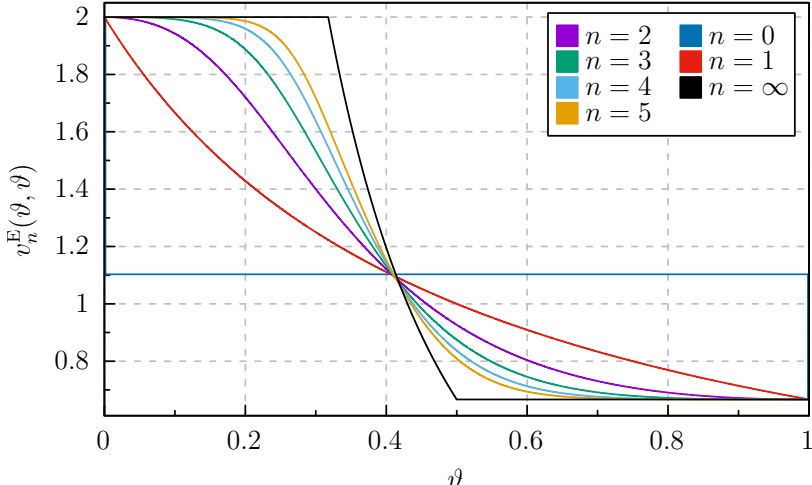

Figure 3: Entanglement velocity (cf. (69)) as a function of the filling $\vartheta$ (note that here we are considering a homogeneous case $\vartheta_{\mathrm{L}} = \vartheta_{\mathrm{R}} = \vartheta$), for different Rényi indices $n$.

which gives the following result for the slope of the von Neumann entropy

$$r(\vartheta_1, \vartheta_2) := \lim_{z \to 1} r_z(\vartheta_1, \vartheta_2) = -\frac{1}{1 + \vartheta_1 + \vartheta_2} \sum_{j=1}^{2} \big( \vartheta_j \log \vartheta_j + (1 - \vartheta_j) \log(1 - \vartheta_j) \big), \qquad (73)$$

or, equivalently

$$\lim_{\substack{t \to \infty \\ |A|/t = \zeta \geq 4}} \frac{S_{A,\mathrm{th}}(t)}{t} = r(\vartheta_{\mathrm{L}}, \vartheta_{\mathrm{R}}) + r(\vartheta_{\mathrm{R}}, \vartheta_{\mathrm{R}}). \qquad (74)$$

## 5 The quasiparticle picture

In the famous work [1], Calabrese and Cardy proposed a simple picture that explains the growth of entanglement in terms of correlated quasiparticles created by the quench. In the simplest formulation one imagines that at $t = 0$ the quench produces pairs of quasiparticles at every point in space and for $t > 0$ they begin to propagate with opposite velocities $\pm v$. Quasiparticles forming each pair are *correlated* or *entangled*, while those in different pairs are uncorrelated. Then, one postulates that, for any time $t$, the entanglement between a given subsystem $A$ and its complement $\bar{A}$ is proportional to the number of correlated pairs shared between $A$ and $\bar{A}$.

Considering a *homogeneous* quench this picture gives the following expression for the Von Neumann entropy

$$S_{A,\mathrm{th}} = \min(4vt, 2|A|)s, \qquad (75)$$

where by $s$ we denoted the contribution to the entanglement of a pair multiplied by the density of pairs. This expression can be immediately generalised to the case of $N_s$ different species of quasiparticles with a dispersion relation parametrised by $\lambda \in [-\Lambda, \Lambda]$

$$S_{A,\mathrm{th}} = \sum_{n=1}^{N_s} \int_{-\Lambda}^{\Lambda} \mathrm{d}\lambda \, \min((v_{n,\lambda} - v_{n,-\lambda})t, |A|) s_{n,\lambda}. \qquad (76)$$

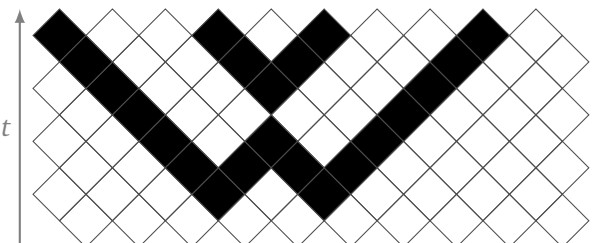

Figure 4: Time-evolution of $|0\rangle^{\otimes 5} \otimes |010\rangle \otimes |0\rangle \otimes |010\rangle \otimes |0\rangle^{\otimes 6}$. Up to the even-odd staggering, the horizontal coordinate of a diamond corresponds to the physical site, the vertical position indicates the time-step (increasing upwards). Black diamonds correspond to $|1\rangle$s and white ones to $|0\rangle$s.

Here we took correlated pairs formed by particles with the same $n$ and opposite $\lambda$s, $v_{n,\lambda}$ is the velocity of the quasiparticle labelled by $(n,\lambda)$ and $s_{n,\lambda}$ the contribution of the pairs labelled by $(n,\lambda)$ to the density of entanglement entropy.

This picture can be generalised to describe inhomogeneous quenches by allowing the contribution to the entanglement of a given pair to depend on the emission point [42] and the quasiparticles to have a curved trajectory [43], namely

$$S_{A,\text{th}} = \sum_{n=1}^{N_s} \int_{-\Lambda}^{\Lambda} d\lambda \int dx \; \chi_A(X_{n,\lambda}(t,x))(1 - \chi_A(X_{n,-\lambda}(t,x)))s_{n,\lambda}(x), \tag{77}$$

where $X_{n,\lambda}(t,x)$ is the position at time $t$ of the quasiparticle $(n,\lambda)$ emitted in $x$ at time 0. Additional refinements accounting for initial states producing $n$-plets of correlated excitations [79,80], and non-unitary non-interacting dynamics [81–83] have also been developed.

Interestingly, due to the simplicity of Rule 54 we can explicitly show that our solvable initial states (25) consist precisely of pairs of oppositely-moving quasiparticles. This can be seen by expressing them in the computational basis

$$\left|\Psi_{\vartheta,\varphi}\right\rangle = \sum_{s_1,s_3,\dots,s_L} \left(e^{i\varphi_1}\sqrt{1-\vartheta}\right)^L \left(e^{i\varphi_2}\sqrt{\frac{\vartheta}{1-\vartheta}}\right)^{s_1+s_2+\dots+s_L} |0s_1 0 s_2 0 s_3 \cdots 0 s_L\rangle. \tag{78}$$

Each of the basis states that enter the above sum at position $2j$ is either $|0\rangle$ or $|1\rangle$. This means that the local configuration around $2j$ is either $|000\rangle$ or $|010\rangle$. We now claim that the first option implies no quasiparticle at position $2j$, while the second one corresponds to a pair of oppositely-moving quasiparticles that have temporarily merged into one site.

This can be appreciated by noting that the time-evolution operator $\mathbb{U}$ is deterministic in the computational basis, therefore each basis state is mapped into exactly one other. In the sequence of bit-strings representing basis states, the freely-moving quasiparticles are given by pairs of consecutive $|1\rangle$ on top of the background of $|0\rangle$. This is conveniently depicted by introducing a staggered zig-zag lattice where even sites are displaced upwards with respect to odd sites, and black and white diamonds correspond to $|1\rangle$ and $|0\rangle$ respectively. Due to the staggering, a freely-moving quasiparticle in a row is graphically represented by one (and not two) black fields. Figure 4 contains an illustration of the time-evolution of a representative basis state included in the sum (78) with $2L = 18$. One observes that each of the local states $|010\rangle$ indeed corresponds to a pair of oppositely moving quasiparticles, and $|000\rangle$ behaves as the empty space. Since in the initial state there cannot be more than one consecutive $|1\rangle$, all quasiparticles appear as pairs.

## 5.1 Von Neumann entropy: exact confirmation of the quasiparticle picture

The quasiparticle picture is believed to apply whenever the system possesses stable quasiparticles [84]. In particular, this is the case for *integrable models* where the "entangling" quasiparticles have been conjectured to coincide with the stable excitations on the stationary state reached after the quench [5]. Using this identification one can make (76) and (77) predictive by computing all the — yet unknown — functions featured in those equations by means of TBA [57,58].

More specifically, let us consider (76) which depends on two unknown functions. The first, $v_n(\lambda)$, is naturally identified with the velocity of the excitations — accessible in TBA [85] — while the second, $s_n(\lambda)$, can be fixed by imposing the equality between the entanglement and the thermodynamic entropy in the stationary state [5]. All this is particularly simple for excitations on the stationary states (35) in Rule 54. Indeed, on these states there is only one species of excitations and $\lambda$ can take only two values ($\lambda \in \{\pm\}$) (for further details see Paper I, the Supplemental Material of Ref. [73], and the review [76]). Therefore, the prediction (76) is effectively of the form (75) with

$$v = v_\vartheta = \frac{2}{1+2\vartheta}, \qquad s = s_\vartheta = -\vartheta \log \vartheta - (1-\vartheta) \log (1-\vartheta), \tag{79}$$

where $\vartheta$ is precisely the filling characterising the Gibbs state (31). Namely, it is written in terms of the chemical potential as in Eq. (33).

We then see that the limits

$$\lim_{\substack{t\to\infty \\ |A|/t=\zeta\geq 4}} \frac{S_{A,\text{th}}}{t} = 4vs, \qquad \lim_{\substack{t\to\infty \\ |A|/t=\zeta\leq 2/3}} \frac{S_{A,\text{th}}}{t} = 2s\zeta, \tag{80}$$

computed with the quasiparticle picture agree with our exact results for all values of $\vartheta$. To the best of our knowledge this result, together with the special case ($\vartheta = 1/2$) presented in Ref. [31], provides the first rigorous confirmation of the quasiparticle picture in the presence of interactions.

The same check can be performed in the case of bipartitioning protocols. In this case, following [42,43], we impose

$$\dot{X}_\pm(x,t) = v_\pm(x,t), \qquad s_+(x) = s_-(x) = s_{\text{L}}\Theta(-x) + s_{\text{R}}\Theta(x), \tag{81}$$

where

$$s_{\text{L/R}} = -\vartheta_{\text{L/R}} \log \vartheta_{\text{L/R}} - \left(1-\vartheta_{\text{L/R}}\right) \log \left(1-\vartheta_{\text{L/R}}\right), \tag{82}$$

and $v_\pm(x,t)$ is the velocity of excitations on the locally quasistationary state at point $(x,t)$ as computed by Generalized Hydrodynamics [40,41]. In particular, using the explicit result for $v_\pm(x,t)$ reported in Paper I we have

$$X_-(x,t) = \begin{cases} x - v_{\text{L}}t, & x < 0, \\ \frac{x}{2}(1+\frac{v_{\text{L}}}{v_{\text{R}}}) - v_{\text{L}}t, & 0 < x \leq 2v_{\text{R}}t, \\ x - v_{\text{R}}t, & x \geq 2v_{\text{R}}t, \end{cases} \tag{83}$$

and

$$X_+(x,t) = \begin{cases} x + v_{\text{L}}t, & x < -2v_{\text{L}}t, \\ \frac{x}{2}(1+\frac{v_{\text{R}}}{v_{\text{L}}}) + v_{\text{R}}t, & -2v_{\text{L}}t \leq x \leq 0, \\ x + v_{\text{R}}t, & x > 0, \end{cases} \tag{84}$$

where we introduced

$$v_{\text{L/R}} = \frac{2}{1+2\vartheta_{\text{L/R}}}. \tag{85}$$

Plugging it into (77), a simple (but tedious) calculation gives

$$
S_{A,\text{th}} = \begin{cases}
\dfrac{2\nu_R\nu_L}{\nu_R+\nu_L}t(s_R+s_L)+2\nu_R t s_R, & \dfrac{|A|}{t} \geq \dfrac{\nu_R(\nu_R+3\nu_L)}{(\nu_R+\nu_L)}, \\[3mm]
\dfrac{2\nu_R\nu_L}{\nu_R+\nu_L}t(s_L-s_R)+2|A|s_R, & \nu_R \leq \dfrac{|A|}{t} \leq \dfrac{\nu_R(\nu_R+3\nu_L)}{(\nu_R+\nu_L)}, \\[3mm]
\dfrac{2\nu_L}{\nu_R+\nu_L}|A|s_L+\dfrac{2\nu_R}{\nu_R+\nu_L}|A|s_R, & \dfrac{|A|}{t} \leq \nu_R,
\end{cases}
\tag{86}
$$

where we took $A = [0, |A|]$. Noting that

$$
\frac{\nu_R(\nu_R+3\nu_L)}{(\nu_R+\nu_L)} < 4, \qquad \nu_R \geq \frac{2}{3},
\tag{87}
$$

we have

$$
\lim_{\substack{t\to\infty \\ |A|/t=\zeta\geq 4}} \frac{S_{A,\text{th}}}{t} = \frac{2\nu_R\nu_L}{\nu_R+\nu_L}(s_R+s_L)+2\nu_R s_R, \qquad \lim_{\substack{t\to\infty \\ |A|/t=\zeta\leq\frac{2}{3}}} \frac{S_{A,\text{th}}}{t} = \frac{2\nu_L}{\nu_R+\nu_L}s_L\zeta + \frac{2\nu_R}{\nu_R+\nu_L}s_R\zeta,
\tag{88}
$$

which, once again, agree with our exact results for all possible values of $\vartheta_{L/R} \in [0,1]$. To the best of our knowledge, this is the first rigorous confirmation of the quasiparticle picture for inhomogeneous quenches.

## 5.2 Rényi Entropies: no consistent quasiparticle description

In non-interacting systems the quasiparticle picture can be directly extended to Rényi entropies with $\alpha \neq 1$. As pointed out in Refs. [56, 86], however, in the presence of interactions this extension becomes far less straightforward. The reason appears to be connected to the fact that $S_A^{(\alpha)}$ have a stronger non-linear dependence on the state compared to the Von Neumann entanglement entropy. This makes it harder to understand which excitations — or better the excitations over which stationary state — are relevant for the quasiparticle picture. As a result, a consistent extension of the quasiparticle picture for Rényi entropies in interacting systems has not yet been found. Here we use our exact results to show that insisting on a quasiparticle description for higher Rényi entropies one has to take excitations over a stationary state with unclear physical meaning. For simplicity, we focus on the homogeneous quench (25) as it contains all the basic elements of our reasoning.

Requiring the validity of the quasiparticle picture we find the following asymptotic formula for the Rényi entropies

$$
S_{A,\text{th}}^{(\alpha)} = \min(4\nu_\alpha(\vartheta)t, 2|A|)s^{(\alpha)}(\vartheta),
\tag{89}
$$

where now $\nu_\alpha(\vartheta)$, $s^\alpha(\vartheta)$ are unknown functions. The density of Rényi entropy "carried" by a quasiparticle pair can be fixed using the expression for the stationary-state Rényi entropy (50), namely

$$
s^{(\alpha)}(\vartheta) = \lim_{\substack{t\to\infty \\ |A|/t=\zeta\leq 2/3}} \frac{S_{A,\text{th}}^{(\alpha)}}{2|A|} = \frac{1}{1-\alpha}\log[\vartheta^\alpha+(1-\vartheta)^\alpha].
\tag{90}
$$

Using now the exact expression for the rate of entanglement spreading (66) we have that the quasiparticle velocity must be given by

$$
\nu_\alpha(\vartheta) = \lim_{\substack{t\to\infty \\ |A|/t=\zeta\geq 4}} \frac{S_{A,\text{th}}^{(\alpha)}}{4ts^{(\alpha)}} = \nu_\alpha^E(\vartheta,\vartheta) = \frac{\log[\lambda_\alpha(\vartheta,\vartheta)]}{\log[\vartheta^\alpha+(1-\vartheta)^\alpha]}.
\tag{91}
$$

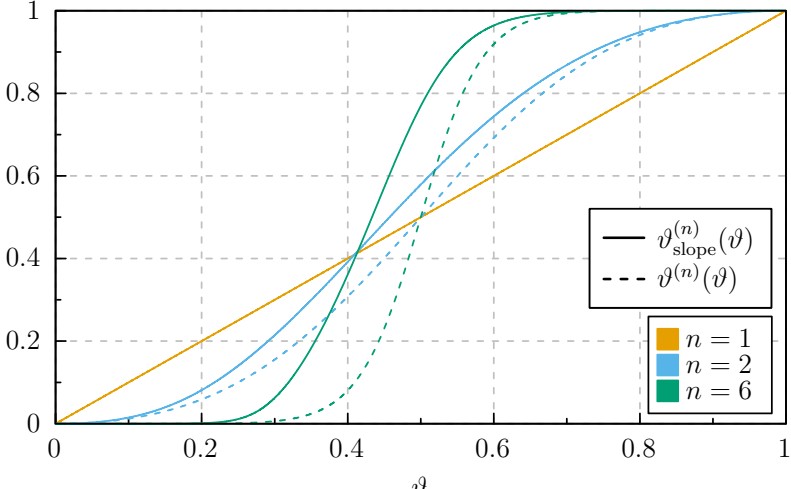

Figure 5: Comparison between the two effective $n$-dependent filling fractions (93) and (94), for a few different choices of $n$. In the case of von Neumann entropy (i.e. for $n = 1$), the effective filling fraction is just $\vartheta$, while for $n \neq 1$ we get two different predictions: $\vartheta_{\text{slope}}^{(n)}$ coming from the renormalised quasiparticle velocity and $\vartheta^{(n)}$ extracted from the stationary state. The two quantities agree for small $\vartheta$ or $1-\vartheta$, while for the intermediate $\vartheta$, the difference between the two increases with $|n-1|$.

Now we note that

$$v_{\alpha \neq 1}(\vartheta) \neq \frac{2}{1 + 2\vartheta}, \tag{92}$$

which means that the quasiparticles *cannot* be though of as excitations on the stationary state (35), i.e. the state describing the expectation values of local observables at infinite times after the quench. Nevertheless, one can interpret $v_\alpha(\vartheta)$ as the velocity excitations over the stationary state with filling

$$\vartheta_{\text{slope}}^{(\alpha)}(\vartheta) = \frac{2 - v_\alpha(\vartheta)}{2 v_\alpha(\vartheta)}. \tag{93}$$

Indeed, since $v_\alpha(\vartheta) \in [2/3, 2]$, Eq. (93) is always in $[0, 1]$ and hence describes a legitimate filling.

The physical meaning of (93) is, however, unclear. In particular, for $\alpha \neq 1$ the filling (93) does *not* coincide with that of the macrostate that describes the stationary value of the $\alpha$ Rényi entropy in TBA [56, 86]. Indeed, in our case the latter has filling (cf. (46))

$$\vartheta^{(\alpha)}(\vartheta) = \frac{\vartheta^\alpha}{\vartheta^\alpha + (1-\vartheta)^\alpha}. \tag{94}$$

Even though (93) and (94) are close for small and large fillings they are different functions of $\vartheta$. See the representative example in Fig. 5.

# 6 Conclusions

In the paper we used a time-channel approach to find exact results for the entanglement dynamics in the quantum cellular automaton Rule 54, which is arguably the simplest example of interacting integrable model. We showed that the entanglement dynamics from a class of solvable initial states is characterised by a certain tensor network and that, remarkably, the latter

can be contracted exactly. We used our results to test the quasiparticle picture for the entanglement spreading in Rule 54. In particular, we confirmed that the quasiparticle picture provides quantitatively accurate predictions for the evolution of the von Neumann entanglement entropy in the presence of interactions, both in homogeneous and inhomogeneous situations. Therefore validating the predictions of both Ref. [5] and Ref. [43]. We also argued that our results seem to exclude a consistent quasiparticle interpretation for the evolution of other Rényi entropies. Indeed, we showed that the potential quasiparticles responsible for the spreading of Rényi entropies cannot be interpreted as excitations on a physically meaningful stationary state.

An interesting direction for future research is to extend the techniques presented here to the study of the various kinds of operator space entanglement [87, 88]. These include the entanglement of local operators, of the reduced density matrix, and of the time evolving operator. The latter is particularly relevant for the questions considered in this paper because it gives access to the "line tension", which is the function needed to obtain quantitative predictions from the membrane picture (see e.g. [89]). It would be interesting to also test these predictions against our results, especially those concerning the Rényi entropies that do not seem to be described by the quasiparticle picture.

## Acknowledgements

We thank Fabian Essler and Maurizio Fagotti for useful discussions and Lorenzo Piroli for collaboration on closely related projects.

**Funding information** This work has been supported by the EPSRC through the grant EP/S020527/1 (KK) and by the Royal Society through the University Research Fellowship No. 201101 (BB).

## A  Proof of Lemma 1

We begin by introducing the following shorthand notation. We call $\{\tilde{M}^{xy}\}_{x,y=1}^{9}$ the set of $4 \times 4$ matrices with matrix elements given by the tensor $\tilde{M}$, i.e.

$$\left[\tilde{M}^{xy}\right]_{ab} := \tilde{M}^{xy}_{ab}. \tag{95}$$

In this new notation the statement of Lemma 1 reads as

$$\left(\tilde{M}^{x_1x_2}\tilde{M}^{y_1y_2}\right) \otimes \left(\tilde{M}^{x_2x_3}\tilde{M}^{y_2y_3}\right) \otimes \cdots \otimes \left(\tilde{M}^{x_kx_1}\tilde{M}^{y_ky_1}\right)$$
$$= \left[\prod_{j=1}^{k} \delta_{x_j,y_j}\right] \left(\tilde{M}^{x_1x_2}\right)^2 \otimes \left(\tilde{M}^{x_2x_3}\right)^2 \otimes \cdots \otimes \left(\tilde{M}^{x_1x_2}\right)^2. \tag{96}$$

Before proving this statement in full generality, let us first consider $k = 1$. In this case one can explicitly evaluate all the products of pairs of matrices $\tilde{M}^{x_1x_1}\tilde{M}^{y_1y_1}$ and realise that the only non-zero combination comes from $x_1 = y_1 = 1$, i.e.

$$\tilde{M}^{x_1x_1}\tilde{M}^{y_1y_1} = \delta_{x_1,0}\delta_{y_1,0}\frac{1}{16}\begin{bmatrix} 1 & 0 & 0 & 0 \\ 0 & 0 & 0 & 0 \\ 0 & 0 & 0 & 0 \\ 0 & 0 & 0 & 0 \end{bmatrix}, \tag{97}$$

and the property (58) holds. For $k = 2$ one can similarly check that the product of two tensors

$$\left(\tilde{M}^{x_1 x_2} \tilde{M}^{y_1 y_2}\right) \otimes \left(\tilde{M}^{x_2 x_1} \tilde{M}^{y_2 y_1}\right) \tag{98}$$

is nonzero only for the following 5 combinations of indices

$$
\begin{array}{c|c|c|c}
x_1 & y_1 & x_2 & y_2 \\
\hline
0 & 0 & 0 & 0 \\
0 & 0 & 8 & 8 \\
8 & 8 & 0 & 0 \\
1 & 1 & 6 & 6 \\
6 & 6 & 1 & 1
\end{array}
\,, \tag{99}
$$

which proves (58) for $k = 2$.

To prove the lemma for general $k$ we show that for any two *different* cycles of indices $(x_1, x_2, x_3, \ldots, x_k, x_1)$ and $(y_1, y_2, \ldots, y_k, y_1)$ at least one of the matrix products $\tilde{M}^{x_j x_{j+1}} \cdot \tilde{M}^{y_j y_{j+1}}$ is 0. This can be demonstrated by defining the $81 \times 81$ adjacency matrix $A$ with elements that are 1 if the two pairs $(x_j, y_j)$ and $(x_{j+1}, y_{j+1})$ are connected by a nonzero matrix product, and 0 otherwise,

$$A_{(x_1 y_1),(x_2 y_2)} := \begin{cases} 1, & \tilde{M}^{x_1 x_2} \cdot \tilde{M}^{y_1 y_2} = 0, \\ 0, & \text{otherwise.} \end{cases} \tag{100}$$

If a pair of cycles $(x_1, x_2, \ldots, x_k, x_1)$ and $(y_1, y_2, \ldots, y_k, y_1)$ gives a nonzero value to the l.h.s. of (58), all the matrix elements of $A$ appearing in the following product have to be 1,

$$A_{(x_1 y_1),(x_2 y_2)} A_{(x_2 y_2),(x_3 y_3)} \cdots A_{(x_k y_k),(x_1 y_1)} = 1, \tag{101}$$

which is one of the contributions to the diagonal matrix element $[A^k]_{(x_1, y_1),(x_1, y_1)}$.

Next, let us define a $9 \times 9$ *reduced* adjacency matrix $\tilde{A}$ that contains only the elements where $x_j$ and $y_j$ are the same

$$\tilde{A}_{x_1, x_2} := A_{(x_1, x_1),(x_2, x_2)}. \tag{102}$$

By explicit diagonalisation of $A$ and $\tilde{A}$ we find that they have the same non-zero eigenvalues, which are the solutions of the following cubic equation

$$x^3 = (x + 1)^2. \tag{103}$$

This implies

$$\text{tr} A^k = \text{tr} \tilde{A}^k, \qquad \forall k, \tag{104}$$

or, in other words, that only non-zero contributions to diagonal elements $[A^k]_{(x_1, y_1),(x_1, y_1)}$ come from elements with the same values of $x_j$ and $y_j$

$$A_{(x_1 y_1),(x_2 y_2)} A_{(x_2 y_2),(x_3 y_3)} \cdots A_{(x_k y_k),(x_1 y_1)} = \delta_{x_1, y_1} \cdots \delta_{x_k, y_k} \tilde{A}_{x_1, x_2} \tilde{A}_{x_2, x_3} \cdots \tilde{A}_{x_k, x_1}. \tag{105}$$

This completes the proof of the lemma.

## B  Proof of Lemma 2

To prove Lemma 2 we rewrite Eq. (63) as

$$p(\lambda, n) = 0, \tag{106}$$

where we defined the polynomial

$$p(\lambda, n) = \lambda^3 + a_{2,n}\lambda^2 + a_{1,n}\lambda + a_{0,n}, \tag{107}$$

with

$$
\begin{aligned}
&a_{0,n} = -\vartheta_1^n \vartheta_2^n, && a_{1,n} = -\left(\vartheta_1^n(1-\vartheta_2)^n + \vartheta_2^n(1-\vartheta_1)^n\right), \\
&a_{2,n} = -(1-\vartheta_1)^n(1-\vartheta_1)^n, && a_{3,n} = 1.
\end{aligned}
\tag{108}
$$

Since $a_{3,n}$ is positive and all $\{a_{j,n}\}_{j=0}^2$ are negative for $\vartheta_1, \vartheta_2 \in (0,1)$, Descartes' rule of signs (see e.g. Ref [90]) implies that Eq. (106) has only one real positive solution which we denote by $\lambda_n(\vartheta_1, \vartheta_2)$. Moreover, since $a_{3,n}$ is positive, we also have

$$p(\lambda, n) > 0, \quad \forall \lambda > \lambda_n(\vartheta_1, \vartheta_2), \tag{109}$$

which implies

$$\lambda^3 > |a_{0,n}| + |a_{1,n}|\lambda + |a_{2,n}|\lambda^2, \quad \forall \lambda > \lambda_n(\vartheta_1, \vartheta_2). \tag{110}$$

Next, we recall that Rouché's Theorem (see e.g. Ref [91]) implies that whenever a polynomial

$$g(\lambda) = \sum_{k=0}^m b_k \lambda^k, \tag{111}$$

has coefficients $b_k \in \mathbb{C}$ fulfilling

$$|b_m|R^m \le \sum_{k=0}^{m-1} |b_k|R^k, \tag{112}$$

for some $R \in \mathbb{R}$, all the solutions to $g(\lambda) = 0$ are contained in the circle of radius $R$. Applying this to (110) we find that all solutions to Eq. (106) are contained in the circle of radius $\lambda_n(\vartheta_1, \vartheta_2)$. To conclude we should prove that the absolute values of the other two solutions to Eq. (63) are strictly smaller than $\lambda_n(\vartheta_1, \vartheta_2)$.

We proceed by contradiction. Let us assume that all solutions to Eq. (106) have the same absolute value. Since the polynomial has real coefficients this means that the solutions are

$$\{\lambda_n(\vartheta_1, \vartheta_2), \ \lambda_n(\vartheta_1, \vartheta_2)e^{i\theta}, \ \lambda_n(\vartheta_1, \vartheta_2)e^{-i\theta}\}, \tag{113}$$

for some $\theta \in \mathbb{R}$. This in turn implies that $p(\lambda, n)$ must coincide with

$$
\begin{aligned}
&(\lambda - \lambda_n(\vartheta_1, \vartheta_2))(\lambda - \lambda_n(\vartheta_1, \vartheta_2)e^{i\theta})(\lambda - \lambda_n(\vartheta_1, \vartheta_2)e^{-i\theta}) \\
&= \lambda^3 - (1 + 2\cos\theta)\lambda_n(\vartheta_1, \vartheta_2)\lambda^2 + (1 + 2\cos\theta)\lambda_n(\vartheta_1, \vartheta_2)^2\lambda - \lambda_n(\vartheta_1, \vartheta_2)^3.
\end{aligned}
\tag{114}
$$

We see that this cannot happen for any $\theta$ because either the coefficient of $\lambda^2$ or that of $\lambda$ are positive, while both $a_{2,n}$ and $a_{1,n}$ are negative. We now assume that there is a single additional solution to Eq. (106) with absolute value equal to $\lambda_n(\vartheta_1, \vartheta_2)$. This implies that the set of solutions reads as

$$\{\lambda_n(\vartheta_1, \vartheta_2), \ -\lambda_n(\vartheta_1, \vartheta_2), \ -\lambda_n(\vartheta_1, \vartheta_2) + c\}, \tag{115}$$

with some $c \in (0, \lambda_n(\vartheta_1, \vartheta_2)]$. Therefore, $p(\lambda, n)$ must coincide with

$$
\begin{aligned}
&(\lambda - \lambda_n(\vartheta_1, \vartheta_2))(\lambda + \lambda_n(\vartheta_1, \vartheta_2))(\lambda + \lambda_n(\vartheta_1, \vartheta_2) - c) \\
&= \lambda^3 + (\lambda_n(\vartheta_1, \vartheta_2) - c)\lambda^2 - \lambda_n(\vartheta_1, \vartheta_2)^2\lambda - \lambda_n(\vartheta_1, \vartheta_2)^2(\lambda_n(\vartheta_1, \vartheta_2) - c).
\end{aligned}
\tag{116}
$$

This is once again impossible because the coefficient of $\lambda^2$ is positive, while $a_{2,n}$ is negative. Therefore the only possibility is that all other solutions to Eq. (106) have absolute value strictly smaller than $\lambda_n(\vartheta_1, \vartheta_2)$. Finally, the explicit expression (64) is found using the general solution of the cubic equation (one can immediately verify that Eq. (64) is indeed real, positive, and fulfils Eq. (63) for all $\vartheta_1, \vartheta_2 \in [0, 1]$).

Let us now move on and prove (70). We begin recalling that $\lambda_z(\vartheta_1, \vartheta_2)$ is obtained replacing $n$ in Eq. (64) with $z \in \mathcal{D} = \{z \in \mathbb{C} \colon \operatorname{Re}[z] > 0\}$ and solves Eq. (106) with $n$ replaced by $z \in \mathcal{D} = \{z \in \mathbb{C} \colon \operatorname{Re}[z] > 0\}$. Next, we observe that

$$1 \geq a_{2,\operatorname{Re}[z]} \geq |a_{2,z}|, \qquad 2 \geq a_{1,\operatorname{Re}[z]} \geq |a_{1,z}|, \qquad 1 \geq a_{0,\operatorname{Re}[z]} \geq |a_{0,z}|. \tag{117}$$

Combining these two facts we find

$$
\begin{aligned}
\lambda_{\operatorname{Re}[z]}(\vartheta_1, \vartheta_2)^3 &= a_{0,\operatorname{Re}[z]} + a_{1,\operatorname{Re}[z]}\lambda_{\operatorname{Re}[z]}(\vartheta_1, \vartheta_2) + a_{2,\operatorname{Re}[z]}\lambda_{\operatorname{Re}[z]}(\vartheta_1, \vartheta_2)^2 \\
&\geq |a_{0,z}| + |a_{1,z}|\lambda_{\operatorname{Re}[z]}(\vartheta_1, \vartheta_2) + |a_{2,z}|\lambda_{\operatorname{Re}[z]}(\vartheta_1, \vartheta_2)^2.
\end{aligned}
\tag{118}
$$

Using again Rouché's Theorem we then have

$$|\lambda_z(\vartheta_1, \vartheta_2)| \leq \lambda_{\operatorname{Re}[z]}(\vartheta_1, \vartheta_2). \tag{119}$$

Finally we observe that

$$3^3 > 1 + 2 \times 3 + 1 \times 3^2 \geq a_{0,\operatorname{Re}[z]} + a_{1,\operatorname{Re}[z]}3 + a_{2,\operatorname{Re}[z]}3^2, \tag{120}$$

which implies

$$3 > \lambda_{\operatorname{Re}[z]}(\vartheta_1, \vartheta_2). \tag{121}$$

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
