# Peer review of "Entanglement dynamics in Rule 54: exact results and quasiparticle picture"

_SciPost Physics, doi:SciPost Phys. 11, 107 (2021)_

## Round 1 · Referee Report · Anonymous (Referee 2) · 2021-5-26

Report

In this work the authors study the quench dynamics in a quantum model for the discrete unitary evolution, the so-called Rule-54. This is an interesting model, which has recently received renewed attention due to the fact that, in some sense, it represents one of the simplest, interacting integrable models. The authors expand the result recently presented in Ref. [31]. They certainly go much beyond what was discussed therein, and the manuscript contains many new and non-trivial results.

The research presented is timely, as it explores an approach to the quench dynamics based on a ``transverse evolution”, which is currently of interest for different groups. In fact, the authors provide a series of exact results which are expected to be appreciated even beyond the field of integrability.

In summary, the main results of the present manuscript are: 1) the derivation of a new family of initial states for which the quench dynamics can be solved exactly; 2) a rigorous derivation of a formula for the asymptotic growth of the Renyi entropies after the quench; 3) the first analytical test of a conjecture due to Alba and Calabrese for the growth of the Von Neumann entanglement entropy; 4) finally, the authors exhibited evidence of the impossibility of establishing a quasi-particle picture for the growth of Renyi entropies.

All these results are of physical significance. In addition, I believe that the paper is very clear and well written.

For the reasons above, I recommend publication.

I have, however, one question for the authors. Although I believe the answer could be of interest, I do not expect them to comment on this on the manuscript.

My question is the following. It is known that Rule 54 can be solved using the Bethe Ansatz method (as shown for instance in Ref. [67]). Therefore, in principle one could expect that the folded transfer matrix could also be diagonalized using Bethe Ansatz. Have the authors explored this direction? How does this relate to the tensor-network approach presented by the authors? In fact, a similar approach could presumably be applied to more general (Floquet) integrable evolutions, such as the Heisenberg XXZ chain, where transfer matrices can be also diagonalized via Bethe Ansatz. Has this approach been explored or do the authors believe there is some fundamental obstacle?

---

## Round 1 · Referee Report · Anonymous (Referee 1) · 2021-8-22

Strengths

1- First exact calculation of R\'enyi entropies in an out-of-equilibrium interacting system.
2- Clear exposition of technical details about the rule 54 and its solution.
3- Rigorous and clearly written.

Weaknesses

1-No major weaknesses.

Report

The paper by Klobas and Bertini study the out-of-equilibrium dynamics of the R\'enyi entropies after a quantum quench in the rule 54 chain, which in recent years emerged as a valuable toy model for generic out-of-equilibrium interacting integrable systems.

This is an excellent paper that deserves publication in Scipost. The most important result of the paper is the calculation of the dynamics of the Renyi entropies in an interacting integrable system. This is a challenging problem. The authors find that while a hydrodynamic description of the dynamics of the Renyi entropies is possible, it is difficult to reconcile it with the standard quasiparticle picture for entanglement spreading, in agreement with previous observations in the literature. I believe that the exact results derived by Klobas and Bertini could be useful to shed light on this problem.

I have only some minor remarks:

1) In the introduction the authors discuss the applicability of the quasiparticle picture to describe the entanglement dynamics. They mention that it has not been applied in systems where the dynamics is not unitary. This is not entirely correct, as it has been extended recently for free fermions and free bosons models in

https://arxiv.org/ct?url=https%3A%2F%2Fdx.doi.org%2F10.1103%2FPhysRevB.103.L020302&v=1136ae07

and

https://arxiv.org/abs/2106.11997

and in a related setting in

https://scipost.org/10.21468/SciPostPhys.7.2.024

2) The authors observe that the dynamics of the Renyi entropies is seemingly not compatible with the quasiparticle picture. As they stress this is in accord with previous observations in the literature (Ref. 73).
However, they should mention that it is still possible to derive the steady
state R\'enyi entropies (as done in Ref. 73) for interacting integrable systems.

3) The the most important results of the paper is the calculation of the Renyi entropies. However, in the introduction they are barely discussed. I would suggest to move part of the discussion on the Renyi entropies in the introduction, for instance mentioning why they are important and what has been done already in the literature (i.e., the calculation of their steady-state value in integrable systems).

---

## Round 2 · Author Response

Response to Referee 1
Q: My question is the following. It is known that Rule 54 can be solved using the Bethe Ansatz method (as shown for instance in Ref. [67]). Therefore, in principle one could expect that the folded transfer matrix could also be diagonalized using Bethe Ansatz. Have the authors explored this direction? How does this relate to the tensor-network approach presented by the authors? In fact, a similar approach could presumably be applied to more general (Floquet) integrable evolutions, such as the Heisenberg XXZ chain, where transfer matrices can be also diagonalized via Bethe Ansatz. Has this approach been explored or do the authors believe there is some fundamental obstacle?
R: We thank the referee for the question, which is indeed very natural and interesting. Approaches based on the Bethe Ansatz diagonalisation of a space transfer matrices have a relative long history in the literature of integrable models. Indeed, this is the main idea of the so called quantum transfer matrix approach
[1] Suzuki, M., and M. Inoue (1987), Prog. Theor. Phys. 78, 787.
[2] Kluemper, A. (1992), Ann. Phys. 504, 540.
[3] Kluemper, A. (1993), Z. Phys. B 91, 507.
Although this approach has mostly been applied to the determination of thermodynamic properties, a program to use it to determine the non-equilibrium dynamics from a class of “solvable” initial states has been proposed in
[4] B. Pozsgay, J. Stat. Mech. (2013) P10028.
[5] L. Piroli, B. Pozsgay, and E. Vernier, J. Stat. Mech. (2017) 023106; Nucl. Phys. B 933, 454 (2018),
where it has been used to determine the time (real and imaginary) evolution Loschmidt echo in the XXZ chain. Note that the latter quantity depends on a non-folded transfer matrix.
The problem of this approach is that, apart from being technically very complicated, it can concretely be applied to determine only a few eigenvalues of the space transfer matrix [5]. In the real-time case these eigenvalues are the leading ones for short times but become sub-leading for large enough times [5]. We believe that this is the main reason why this approach has not yet successfully been applied to study the dynamics of local observables and entanglement, which involve more complicated folded transfer matrices.
We believe that a similar problem would arise for Rule 54: even though in principle one can determine the spectrum of W using this approach (in fact, to apply the standard quantum transfer matrix approach one would need a ABA treatment of Rule 54 which is currently not fully developed) this is probably going to be more complicated than the treatment employed here (another advantage of the technique used here is that it is in principle independent of integrability). However, as the leading eigenvectors are ultimately uniquely defined and do not depend on the method used to obtain them, it would be very interesting to find them using integrability methods. First it would be interesting to see whether the “solvable” initial states which we find here correspond to the Bethe Ansatz ones (we believe that this is the case because our solvable states produce pairs of quasiparticles as the solvable states in Bethe Ansatz). Second it would be interesting to understand whether a Bethe Ansatz approach provides an explanation
for the remarkably simple form of the fixed points in Rule 54.
This question of the referee made us realise that it is appropriate to mention such integrability-based approaches to study the space transfer matrix. However, we decided to include a brief discussion about them in Paper I, where we survey a number of different approaches based on the “space-like” evolution.
Response to Referee 2
Q: In the introduction the authors discuss the applicability of the quasiparticle picture to describe the entanglement dynamics. They mention that it has not been applied in systems where the dynamics is not unitary. This is not entirely correct, as it has been extended
recently for free fermions and free bosons models in
https://doi.org/10.1103/PhysRevB.103.L020302
and
https://arxiv.org/abs/2106.11997
and in a related setting in
https://scipost.org/10.21468/SciPostPhys.7.2.024
R: We thank the referee for pointing out these relevant references, which we added to our bibliography.
We decided, however, not to quote them in the introduction, where the discussion is about the physical behaviour, i.e. whether or not the entanglement entropies grow linearly, rather than the precise technique used to describe it. Indeed, in the scenarios discussed in the above references the entropies either saturate or grow logarithmically in time as opposed to the standard linear growth that one observes in the unitary case.
In our view it is more appropriate to mention the above references in Sec. 5, when discussing the quasiparticle picture.
Q: The authors observe that the dynamics of the Rényi entropies is seemingly not compatible with the quasiparticle picture. As they stress this is in accord with previous observations in the literature (Ref. 73). However, they should mention that it is still possible to derive the steady state Rényi entropies (as done in Ref. 73) for interacting integrable systems.
R: We thank the referee for pointing out this. We are now mentioning this point both in Sec. 2 (after equation 2.6) and in Sec. 4 (after equation 4.9).
Q: The most important results of the paper is the calculation of the Renyi entropies. However, in the introduction they are barely discussed. I would suggest to move part of the discussion on the Renyi entropies in the introduction, for instance mentioning why they are important and what has been done already in the literature (i.e., the calculation of their steady-state value in integrable systems).
R: The referee is indeed right in saying that the calculation of the time evolution of Rényi entropies is our main result. However, we prefer to leave the discussion in the introduction on a slightly more general level (talking about entanglement rather than its measures). This is for two main reasons: (i) we believe that talking about entanglement is more significant from the physical point of view; (ii) in many cases the same techniques can be applied to the calculation of several entanglement measures, not only the Renyi entropies.
We nevertheless agree with the referee that it is useful to slightly expand the discussion about Rényi entropies, accounting for the previous literature on the matter. Especially in the case of interacting integrable models. Therefore we expanded the discussion after Eq. 2.6.

---

## Round 2 · List of Changes

All changes are denoted in red in the manuscript.

---

## Editorial Decision

published